# Evaluation of Fruquintinib in the Continuum of Care of Patients with Colorectal Cancer

**DOI:** 10.3390/ijms24065840

**Published:** 2023-03-19

**Authors:** Daniele Lavacchi, Giandomenico Roviello, Alessia Guidolin, Silvia Romano, Jacopo Venturini, Enrico Caliman, Agnese Vannini, Elisa Giommoni, Elisa Pellegrini, Marco Brugia, Serena Pillozzi, Lorenzo Antonuzzo

**Affiliations:** 1Clinical Oncology Unit, Careggi University Hospital, 50134 Florence, Italy; 2Department of Health Science, University of Florence, 50139 Florence, Italy; 3Department of Experimental and Clinical Medicine, University of Florence, 50134 Florence, Italy; 4Medical Oncology Unit, Careggi University Hospital, 50134 Florence, Italy

**Keywords:** fruquintinib, CRC, VEGFR2, tyrosin kinase inhibitor

## Abstract

The management of patients with metastatic colorectal cancer (mCRC) has the continuum of care as the treatment paradigm. To date, trifluridine/tipiracil, a biochemically modulated fluoropyrimidine, and regorafenib, a multi-kinase inhibitor, remain the main options for the majority of patients who progressed to standard doublet- or triplet-based chemotherapies, although a tailored approach could be indicated in certain circumstances. Being highly selective for vascular endothelial growth factor receptor (VEGFR)-1, -2 and -3, fruquintinib demonstrated a strong anti-tumor activity in preclinical models and received approval from China’s National Medical Products Administration (NMPA) in 2018 for the treatment of patients with chemo-refractory mCRC. The approval was based on the results of the phase III FRESCO trial. Then, in order to overcome geographic differences in clinical practice, the FRESCO-2 trial was conducted in the US, Europe, Japan, and Australia. In a heavily pretreated patient population, the study met its primary endpoint, demonstrating an advantage of fruquintinib over a placebo in overall survival (OS). Here, we review the clinical development of fruquintinib and its perspectives in gastrointestinal cancers. Then, we discuss the introduction of fruquintinib in the continuum of care of CRC paying special attention to unmet needs, including the identification of cross-resistant and potentially susceptible populations, evaluation of radiological response, and identification of novel biomarkers of clinical benefit.

## 1. Introduction

Colorectal cancer (CRC) is the third most common tumor with approximately 1,931,590 cases annually, and the second leading cause of cancer-related death worldwide [1]. Nearly 15–30% of patients are diagnosed with advanced disease, while 20–50% of cases with resectable disease will develop metachronous metastases. The 5-year survival rate for the metastatic stage is approximately 14% [1,2].

The treatment paradigm for CRC is nowadays moving towards a tailored approach based on clinical and molecular characteristics. The combination of chemotherapy doublets or triplets with monoclonal antibodies (moAbs) remains the standard of treatment for the vast majority of patients with microsatellite stable (MSS) metastatic CRC (mCRC). The choice of the moAb lies between drugs directed against the epidermal growth factor receptor (EGFR) and the vascular endothelial growth factor receptor (VEGFR), according to the patient characteristics, tumor molecular profile, and primary tumor location [3,4,5,6,7,8]. In contrast, mismatch repair deficient (dMMR) CRC patients are a highly selected subgroup who have been shown to receive a remarkable benefit from immune checkpoint inhibitors as the chemo-free treatment strategy [9,10]. Being a key process for tumor growth and metastasis, angiogenesis has been considered a therapeutic target in the continuum of care of mCRC [11,12]. During rapid cell replication, hypoxic conditions trigger the activation of the hypoxia-inducible factor 1-alpha (HIF1A), which induces the transcription of more than 60 genes, such as the pro-angiogenic factor VEGF (vascular endothelial growth factor), thus promoting oxygen delivery and cell survival [13]. The VEGF/VEGFR axis is composed of multiple ligands (i.e., VEGF-A, -B, -C, -D, -E, and placental growth factor) and tyrosine kinase receptors (VEGFR1, 2, and 3) with different binding affinities and functions [14].

Clinical studies have shown that anti-angiogenic drugs improve survival in patients with mCRC [15]. Bevacizumab, a moAb directed against the VEGF-A ligand, is the first anti-angiogenic drug approved in combination with cytotoxic chemotherapy as the first-line treatment in mCRC patients. From the results of the pivotal phase III AVF2107 trial [16], several other clinical trials investigated the effects of bevacizumab across various treatment lines, extending the indications for the second-line or beyond-progression therapy [17,18]. Among the resistance mechanisms to the anti-VEGF-A blockade, a decrease in VEGF-A and an increase in PDGF, VEGF-C, and VEGF-D levels after bevacizumab treatment has been reported by Hayashi et al. [19,20,21]. This suggestion paved the way for the development of other agents able to target multiple signaling pathways simultaneously. Displaying a high affinity to VEGF-A, VEGF-B, and placental growth factor, aflibercept showed a statistically significant overall survival (OS) improvement in the VELOUR study and in real-world datasets. The benefit has been observed both in bevacizumab-pretreated patients and in bevacizumab-naïve patients, thus making the drug an alternative second-line therapy [22,23].

In patients with mCRC who are refractory to these treatments, regorafenib, a multi-kinase inhibitor, and trifuridine/tipiracil, a biochemically modulated fluoropyrimidine, have been shown to improve OS in the randomized CORRECT and RECOURSE trials, respectively [24,25]. Regorafenib is an oral multi-kinase (anti-VEGFR1/3, PDGFR, and FGFR) and mutant oncogenic kinase (KIT, RET, and BRAF) inhibitor with antiangiogenic proprieties. Its efficacy in heavily pretreated patients may be due to the broad spectrum of anti-kinase activity, which conversely, may imply a higher incidence of adverse events (AE). However, a narrower range of targets might minimize off-target toxicities and improve the clinical outcome due to a higher drug exposure at the maximum tolerated dose (MTD) [26,27,28,29,30,31].

Hence, fruquintinib, a highly selective tyrosine kinase inhibitor targeting VEGFR1, 2, and 3, was developed based on the strong anti-tumor activity shown in preclinical models [30,31]. Then, based on the results of the phase III FRESCO study, fruquintinib received its first approval from China’s National Medical Products Administration (NMPA) on 4 September 2018 for the treatment of mCRC in which at least two prior systemic therapies had failed [32,33]. Furthermore, given the geographic differences in clinical practice, the global FRESCO-2 trial was conducted. The study was positive and met its primary endpoint: fruquintinib was associated with an improvement in OS compared with the placebo [34]. The purpose of this review is to discuss the current data and future perspectives of fruquintinib for the treatment of patients with chemo-refractory mCRC.

## 2. Pharmacodynamic Properties

Fruquintinib (6-[6,7-dimethoxyquinazolin-4-yloxy]-N, 2-dimethylbenzofuran-3-carboxamide) is a new generation potent tyrosine kinase inhibitor of VEGFR1, 2, and 3 [30] (Figure 1). This bond prevents VEGFR conformational change and dimerization and consequently, the phosphorylation of the intracellular kinase domain, which would trigger downstream signaling cascades, such as the PI3K/AKT, PKC, RAF/RAS, and ERK pathways [35,36]. VEGFR2 is a crucial member of the VEGFR family, being deeply involved in pro-angiogenic processes, whereas VEGFR1 seems to act as a negative regulator of the R2 signaling [35,37]. VEGFR3 is only expressed on lymphatic vessels and endothelial cells, thereby, promoting lymphangiogenesis and lymph node metastasis [38,39].

Fruquintinib has shown optimal antitumor activity, both in vitro and in vivo, in pre-clinical models [32]. In vitro studies were conducted on human umbilical vein (HUVEC) and lymphatic endothelial cells (HLEC) to evaluate both the angiogenic VEGFR2 and the lymphangiogenic VEGFR3 pathways, towards which fruquintinib demonstrated an equal inhibitory potential. In vitro, fruquintinib displayed anti-angiogenetic properties, suppressing endothelial cell proliferation and tubule sprouting in a dose-dependent fashion. Its kinase selectivity was tested against a panel of 253 kinases. A potent inhibition of VEGFR1, 2, and 3 was shown, with IC50s of 33 nmol/L, 35 nmol/L, and 0.5 nmol/L, respectively. A weak activity (IC50 values of 128–458 nmol/L) against RET, FGFR1, and c-KIT kinases has also been reported. The potent in vitro activity against VEGFR was then confirmed in vivo following administration in multiple human tumor xenograft murine models of colon, renal, gastric, and lung cancer. A near complete (>85%) inhibition of the VEGFR2 was obtained for at least 8 h after a single oral dose of fruquintinib at 2.5 mg/kg. Furthermore, the association with chemotherapeutic agents has been investigated. Enhanced antitumor activities were observed when fruquintinib was administered in combination with docetaxel and oxaliplatin in gastric cancer and colon cancer patient-derived xenograft (PDX) models, respectively, resulting in approximately a 30% decrease in tumor growth inhibition (TGI) rate. Other drug combinations in xenograft models have been evaluated due to the fact that certain cell lines (i.e., renal cancer models) showed scarce TGIs with fruquintinib monotherapy [30]. Interestingly, the coadministration of fruquintinib and the c-MET inhibitor savolitinib or the tyrosine kinase inhibitor geftitinib produced a marked reduction in tumor growth in preclinical models [40]. Furthermore, the influence of anti-VEGF therapy on the tumor immune microenvironment was examined on CRC allograft tumor models. Interestingly, low doses of fruquintinib combined with sintilimab, an anti-programmed death-1 (PD-1), seemed capable of reprogramming the immune response. A reduced angiogenesis, together with enhanced infiltration of CD8+T cells and reduced ratios of immunosuppressive cells, such as myeloid-derived suppressor cells (MDSCs) and macrophages, was described. Of note, the combination of anti-PD-1 and anti-VEGF achieved effective responses in patients with refractory MSS mCRC, suggesting a relevant synergistic effect [41].

## 3. Clinical Development

Here we present the main trials that were crucial for the clinical development of fruquintinib in patients with chemo-refractory mCRC.

### 3.1. Phase 1–2

Recommended phase II dose (5 mg once daily for 3 weeks on and 1 week off) was determined from a Phase I trial, involving 40 Chinese patients with different tumor types (i.e., CRC, lung cancer, breast cancer, gastric cancer, melanoma). The study design included several dose cohorts: 1–6 mg on the continuous regimen and 5–6 mg for 3 weeks on and 1 week off regimen. Two patients experienced grade (G) 3 hand–foot skin (HFS) reaction as dose-limiting toxicity leading to treatment discontinuation in the 6-mg cohort. In the 5-mg cohort, after the enrolment of an additional three patients, one G3 symptomatic thrombocytopenia and one G3 HFS reaction were observed. Therefore, 4 mg was determined as the MTD for the continuous regimen. After the expansion of the 4-mg cohort, no other DLT was reported. Considering the AE time to onset, the dose level of 5 mg for the 3 weeks on and 1 week off regimen was selected. None of the first eight patients included in this cohort had a DLT. In contrast, at the 6-mg dose level, one patient experienced G3 fatigue. Overall, HFS reaction, hypertension, and thrombocytopenia were the most commonly reported AEs. Serious AEs were observed in 7.5% of cases. Among all G AEs, an HFS reaction was observed in 77.5%, hypertension in 42.5%, proteinuria in 47.5%, and a G1 TSH increase in 67.5%. Among the patients evaluated for response, the overall response rate (ORR) was 41.1%, and disease control was obtained in 82.3%. Three patients with mCRC obtained a partial response (PR) and two young women with chemo-refractory lung cancer and breast cancer had a long-term PR (PR duration of 12 months and 13.2 months, respectively). Pharmacokinetic analyses revealed a high plasma exposure after oral administration and long half-life that supported the 3 weeks on and 1 week off regimen. In fact, the steady state was reached after two weeks of treatment and maintained in the third week. A gradual decrease until complete elimination was observed in the treatment-free week [42].

An open-label, single-arm phase Ib trial was conducted in two hospitals in China (NCT01975077), between December 2012 and January 2014. The study included patients with mCRC who progressed after at least two previous treatment lines, including fluoropyrimidine, oxaliplatin, or irinotecan-based regimens. In the extension stage of the study, the regimen that was chosen for further development was fruquintinib 5 mg daily for 3 weeks on and 1 week off. The primary endpoint was progression-free survival (PFS). Forty-two patients aged between 33 and 70, and with good ECOG performance status (0–1), were enrolled. The vast majority of patients (88.1%) received more than three previous treatment lines. The median PFS was 5.8 months (95% CI 4.01–7.60). Median OS was 8.9 months (95% CI 7.53–15.53). ORR and disease control rate (DCR) were 9.5% and 76.2%, respectively. Although all the patients included in the study developed treatment-emergent AEs, toxicities that led to permanent discontinuation in 5 patients were chest pain, pancreatitis, hemoptysis, proteinuria, and skin toxicity. The most commonly reported G3-4 treatment-emergent AEs were hypertension in 21.4%, diarrhea in 9.5%, HFS reaction in 9.5%, and serum sodium decrease in 7.1%. About half of the patients (47.6%) required a dose reduction or interruption. One toxic death due to hemoptysis was reported [43,44].

A randomized, double-blind, multicenter phase II trial was conducted in eight hospitals in China (NTC02196688), between April 2014 and August 2014. Seventy-one patients were randomly assigned (2:1) to receive fruquintinib plus best supportive care (BSC) or placebo plus BSC. Baseline characteristics were well balanced across the two treatment arms. Inclusion criteria were similar to the phase Ib study. The vast majority of patients (71–74%) was previously treated with at least three treatment lines, but only 29–32% received prior anti-VEGF agent. Treatment duration was 3.2 months in the fruquintinib group and 0.8 months in the placebo group. The study met the primary endpoint: fruquintinib was associated with a significantly improved PFS than placebo (median PFS 4.32 months vs. 0.99 months; HR 0.30; 95% CI 0.15–0.59; *p* < 0.001). Patients treated with fruquintinib also had a significantly higher DCR compared to placebo (68.1% vs. 20.8%, *p* < 0.001). However, there were no significant differences in terms of ORR and OS between the two groups, although a trend toward a better OS was observed in the fruquintinib arm (Table 1). Overall, a higher incidence of AEs was observed in the fruquintinib group than in the placebo group (93.6% vs. 58.3%). The most common G3-4 treatment-emergent AEs in the fruquintinib group were hypertension (28.9%) and HFS reaction (14.9%). In the experimental arm, 25.5% of patients experienced a serious AE. Dose modification was 61.7% in the fruquintinib group, and 29.2% in the placebo group, and interruption rates due to AEs were 34% and 16.7%, respectively [43].

### 3.2. Phase 3

The phase III FRESCO (Fruquintinib Efficacy and Safety in 3+ Line Colorectal Cancer Patients) trial was a randomized, double-blind, placebo-controlled, multicenter study (28 hospitals in China). From December 2014 to June 2017, 416 patients were randomized (2:1) to receive fruquintinib plus BSC or placebo plus BSC. The study population included patients who had mCRC and experienced progressive disease (PD) after two standard lines of treatment containing fluoropyrimidine, irinotecan, and oxaliplatin, an anti-VEGF therapy and, if wild-type RAS, an anti-EGFR moAb. Patients who received VEGFR inhibitors (e.g., regorafenib, ramucirumab, or apatinib) were excluded. Anti-VEGF therapy and KRAS mutational status were stratification factors. Disease characteristics were well balanced in both treatment arms, with a high proportion of patients having multi-organ metastasis (95.3% in the fruquintinib arm and 97.1% in the placebo arm) and the left colon as the primary tumor location (77.0% and 83.3%, respectively). Among patients in the fruquintinib arm, 30.2% previously received bevacizumab and/or aflibercept, and 14.4% received cetuximab. Fruquintinib 5 mg per os, administered with the 3 weeks on and 1 week off scheme, significantly improved median OS compared with placebo meeting the primary endpoint (median OS: 9.3 months [95% CI: 8.2–10.5] vs. 6.6 months [95% CI: 5.9–8.1]; HR 0.65; 95% CI: 0.51–0.83; *p* < 0.001). The OS benefit was observed across nearly all subgroups, including patients who previously received more than three treatment lines. Among the key secondary endpoints, median PFS was also significantly longer in the fruquintinib arm compared with the placebo arm (3.71 months vs. 1.84; HR 0.26; 95% CI, 0.21–0.34; *p* < 0.001); ORR (4.7% vs. 0%, respectively) and DCR (62.2% vs. 12.3%, respectively) were also higher in the experimental arm. At the time of PD, 45.2% of patients received subsequent treatments (42.4% in the fruquintinib arm and 50.7% in the placebo arm). G3-4 AEs were experienced by 61.2% of patients receiving fruquintinib, including hypertension in 21.2%, HFS reaction in 10.8%, and proteinuria in 3.2%. Serious AEs were reported in 15.5% of cases receiving fruquintinib. Eleven patients reported G5 AEs (nine in the fruquintinib arm and two in the placebo arm), including cases of gastrointestinal hemorrhage, stroke, and hemoptysis. Dose discontinuation was reported in 15.1% in the fruquintinib arm, and treatment interruption or dose reduction in 47.1%.

When this study was conducted in China, the standard of care for the treatment of mCRC differed from the current standard of care in Western countries, in which only one-third of patients received previous treatment with anti-VEGF or anti-EGFR antibodies. In addition, neither third-line treatment with trifluridine/tipiracil nor regorafenib was available in China at that time, so it was not possible to translate the results into the context of the Western population. An additional limitation was the absence of microsatellite instability status, which may influence prognosis and response to further treatments [33].

FRESCO-2 is a global, randomized, double-blind, placebo-controlled, multicenter, phase III trial comparing fruquintinib plus BSC with placebo plus BSC in patients with mCRC that had experienced PD, despite two or more prior chemotherapy regimens with or without prior anti-VEGF or anti-EGFR agents. From September 2020 to December 2021, 687 patients from the US, Europe, Japan, and Australia were randomized in a 2:1 ratio to receive fruquintinib 5 mg per os, using the 3 weeks on and 1 week off scheme or a placebo. Treatment continued until progression or toxicity. The study included patients that received trifluridine/tipiracil (52.1% in the experimental arm) or regorafenib (8.7%) or both (39.3%), immunotherapeutic agents, or BRAF inhibitors, if indicated. Among exclusion criteria there were brain metastases and/or invasion of large vascular structures. The vast majority of patients received >3 treatment lines (72%), including an anti-VEGF agent in approximately 96% of cases. The primary endpoint was met: fruquintinib significantly prolonged OS compared with placebo (7.4 months vs. 4.8; HR = 0.66; 95% CI, 0.55–0.80; *p* < 0.001). The study also met key secondary endpoints, such as PFS (3.7 months vs. 1.8; HR = 0.32; 95% CI: 0.27, 0.39; *p* < 0.001), confirmed ORR (1.5% vs. 0%; *p* = 0.059), and DCR (55.5% vs. 16.1%; *p* < 0.001) (Table 1). OS subgroup analysis confirmed the benefit received from fruquintinib across all subgroups, including patients who received both trifluridine/tipiracil and regorafenib (HR 0.6). In the safety analysis, G ≥ 3 AEs were observed in a slightly higher percentage in the fruquintinib arm than in the placebo arm (62.7% and 50.4%, respectively). The most commonly reported G3-4 AEs in the fruquintinib arm were hypertension (13.6%), asthenia (7.7%), and HFS reaction (6.4%) that led to dose reduction in 3.7%, 3.5%, and 5.3% of cases, respectively. In the experimental arm, dose interruption, reduction, and discontinuation rates due to treatment-emergent AEs were 54.2%, 24.1%, and 20.4%, respectively [34].

## 4. Brief Discussion

The management of patients with mCRC has the continuum of care as its paradigm. As early as 2004, Grothey and colleagues showed that mCRC patients benefited from receiving all available active agents for which they were candidates. At that time, only 5FU, irinotecan, and oxaliplatin were available agents, and patient survival was closely related to the possibility of receiving all three drugs (*p* = 0.0008) [45]. The concept of the continuum of care was then confirmed over the years. The introduction of moAbs, multi-kinase inhibitors, and new fluoropyrimidines has led to a median survival of over two years. [24,25,46,47] To date, a comprehensive treatment strategy with the integration of sequential chemotherapies, biological agents, surgery, local treatments, off-treatment periods, and best supportive care is a prerequisite to obtaining excellent outcomes in selected patients.

In the FRESCO-2 trial, fruquintinib also demonstrated an improvement in OS in a heavily pretreated patient population since 72.9% of patients received more than three previous treatment lines. In the fruquintinib arm, 52.1% of patients previously received trifluridine/tipiracil, and 39.3% previously received both trifluridine/tipiracil and regorafenib. However, a treatment benefit was demonstrated in both these subgroups (OS HR 0.72 and 0.60, respectively), highlighting the possibility of using fruquintinib in a third- or subsequent line of treatment [34]. These data also suggest that there is no complete refractoriness in patients previously treated with regorafenib, suggesting the opportunity to analyze further subgroups, which are the cross-resistant and potentially susceptible populations.

Every effort should be made to identify clinical and molecular biomarkers predictive of response. Potential factors associated with treatment response to antiangiogenics and multitarget agents included LDH, hERG1/aHIF-2α expression, the circulating angiopoietin-2 level, but results should be confirmed in prospective studies [27,48,49,50,51]. An exploratory analysis from the CORRECT trial showed an association between HFS reaction and survival benefit from regorafenib, and a similar differential benefit was observed for fruquintinib in a post-hoc analysis of the FRESCO trial [52,53].

Another crucial issue to be addressed is whether the radiological response, according to RECIST1.1 criteria, is appropriate to evaluate patients receiving fruquintinib. A post-hoc analysis from the CORRECT trial evaluated the cavitation of lung metastases as an early biomarker of treatment response. The authors showed that 34.3% of patients in the regorafenib arm developed cavitation of lung metastases, and the majority of patients who had cavitation at baseline experienced an increase in cavitation at week 8. This pattern was associated with PFS and a higher rate of disease control. DCR was 69.7% in patients with cavitation at week 8 and 42.3% in those without cavitation (*p* = 0.01). Albeit limited in size, patients with an increase in pre-existent cavitation also had a higher disease control as compared with those without an increase in cavitation (*p* = 0.004) [54]. Although RECIST 1.1 remains the main predictor of treatment benefit, being closely associated with survival, a comprehensive evaluation of other radiological characteristics could enrich the interpretation of treatment response in patients receiving anti-angiogenetic agents. To date, little is known about the specific pattern of the response to fruquintinib, but less explored radiological parameters, such as cavitation in lung metastases, should be extensively studied as early biomarkers of clinical benefit.

## 5. Perspectives

To date, several studies are exploring the activity of fruquintinib in gastrointestinal cancers (Table 2). In mCRC, some phase II studies are evaluating the combination of fruquintinib and FOLFOX/FOLFIRI as first- or second-line treatments (NCT05004441, NCT05634590). In addition, fruquintinib combined with capecitabine is under evaluation as a first-line treatment in a phase II trial, including patients unsuitable for intravenous chemotherapy. A phase II trial aims to evaluate the efficacy and safety of fruquintinib plus FOLFIRI vs. bevacizumab plus FOLFIRI as a second-line treatment in mCRC (NCT05555901). The study will enroll approximately 272 patients in several Chinese centers. Similarly, the combination of fruquintinib plus FOLFIRI as a second-line treatment is under evaluation in a single-center phase Ib/II trial, including patients with RAS-mutated mCRC (NCT05522738). Another phase II study aims to evaluate the efficacy and safety of the combination of fruquintinib and trifluridine/tipiracil as a third-line treatment in mCRC (NCT05004831). The best sequence (FOLFIRI bevacizumab followed by fruquintinib or vice versa) is under evaluation in a phase II trial that aims to enroll approximately 134 participants (NCT05447715). The combination of fruquintinib with sintilimab, an anti-PD-1 moAb, is under evaluation in a phase II study for chemo-refractory mCRC (NCT04695470). Several phase II studies are evaluating fruquintinib as a maintenance therapy following first-line treatment for mCRC patients (NCT04296019, NCT05016869, NCT05451719, NCT04733963, NCT05659290). In chemo-refractory mCRC, a phase II trial is exploring the addition of raltitrexed to fruquintinib vs. single-agent fruquintinib (NCT04582981). In a similar setting, a phase II study aims to evaluate the combination of fruquintinib and camrelizumab, an anti-PD-1 antibody (NCT04866862). Two phase II trials are exploring the combination of radiotherapy, fruquintinib, and an anti-PD-1 agent (NCT05292417, NCT04948034). A dose-escalating phase Ib trial is evaluating the safety of GB226, an anti-PD-1 moAb, in combination with fruquintinib (NCT03977090). Three phase II trials are investigating fruquintinib in combination with hepatic arterial infusion chemotherapy in mCRC (NCT05406206, NCT05511051, NCT05435313). In HER2-positive/-mutated mCRC patients, a single-arm study will explore the combination of fruquintinib and disitamab vedotin, an antibody-drug conjugate directed against HER2 with a cleavable linker to the monomethyl auristatin E, after standard treatment failure (NCT05661357). A phase II study will evaluate fruquintinib combined with FOLFOX and radiotherapy in locally advanced rectal cancer (NCT05575635). Among non-CRC cancers, initial positive results have been announced for the combination of fruquintinib and paclitaxel in a second-line treatment of advanced gastric cancer (GC): the study (NCT03223376—FRUTIGA trial) met the primary endpoint (PFS). The combination of fruquintinib and SOX is under evaluation in a phase II trial as a neoadjuvant treatment for patients with locally advanced GC (NCT05122091). In a first-line treatment of advanced GC, a phase Ib/II trial is evaluating the combination of fruquintinib, toripalimab, and SOX (NCT05024812). Similarly, a phase II study is evaluating fruquintinib, sintilimab, and SOX as conversion therapy (NCT05177068). In a second-line treatment of advanced GC, a phase II trial will evaluate fruquintinib plus irinotecan (NCT05643677), and another phase II trial will evaluate fruquintinib plus sintilimab (NCT05625737). In HER2-positive advanced GC, the combination of fruquintinib and disitamab vedotin is being studied in a phase II trial in previously treated patients (NCT05241899). A phase II trial is evaluating fruquintinib plus tislelizumab, an anti-PD-1 moAb, in chemo-refractory GC, CRC, and non-small cell lung cancer (NSCLC) (NCT04716634). The same drug combination will be evaluated in a phase II trial, including patients with MSS locally advanced rectal cancer with a high immune score (NCT04989855). In advanced pancreatic cancer, fruquintinib is under evaluation in phase II trials as a first-line treatment in combination with nabpaclitaxel and gemcitabine (NCT05168527) or as a single-agent in third- or further-line treatments (NCT05257122). In esophageal squamous cell carcinoma, a phase II trial is evaluating the combination of fruquintinib and S-1 after standard treatment failure (NCT05636150).

In conclusion, fruquintinib is an optimal candidate to be incorporated into the individually optimized treatment plan for patients with mCRC. Novel drug combinations are being studied in gastrointestinal cancers in order to improve survival and quality of life.

## Figures and Tables

**Figure 1 ijms-24-05840-f001:**
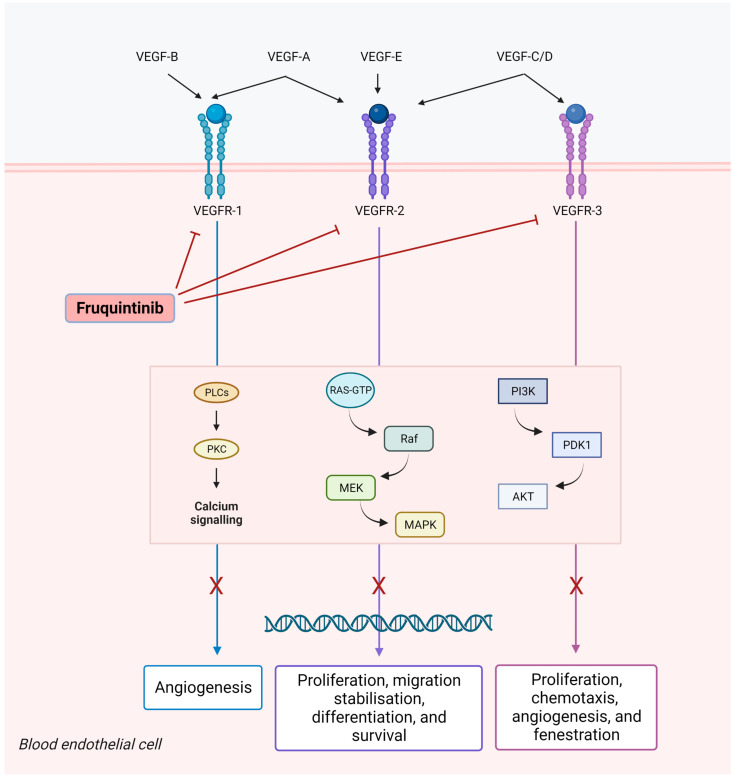
Fruquintinib inhibits vascular endothelial growth factor (VEGF)-induced phosphorylation of VEGF receptors 1, 2, and 3 and related signaling pathways. This may result in the inhibition of migration, proliferation, and survival of endothelial cells, micro-vessel formation, the inhibition of tumor cell proliferation, and tumor cell death.

**Table 1 ijms-24-05840-t001:** Summary of the main trials of fruquintinib in colorectal cancer.

Trial	Phase	Therapy-Line	Arms	Number of Patients with CRC	Primary End-Point	ORR (%)	DCR (%)	PFS (Months)	OS (Months)
NCT01975077 [43]	I/II	Third-line	- Fruquintinib 5 mg PO, QD (3 weeks on, 1 week off)	42	PFS	9.5	76.2	5.8	8.88
NCT02196688 [43]	II	Third-line	- Fruquintinib 5 mg PO, QD (3 weeks on, 1 week off) - Placebo	71	PFS	2.1 vs. 0 (*p* = 0.45)	68.1 vs. 20.8 (*p* < 0.001)	4.73 vs. 0.99 (*p* < 0.001)	7.72 vs. 5.52 (*p* = 0.29)
FRESCO NCT02314819 [33]	III	Third-line	- Fruquintinib 5 mg PO, QD (3 weeks on, 1 week off) - Placebo	416	OS	4.7 vs. 0 (*p* = 0.01)	62.3 vs. 12.3 (*p* <0.001)	3.7 vs. 1.8 (*p* < 0.001)	9.3 vs. 6.6 (*p* < 0.001)
FRESCO-2 NCT04322539 [34]	III	Third-line	- Fruquintinib 5 mg PO, QD (3 weeks on, 1 week off) - Placebo	687	OS	1.5 vs. 0 (*p* = 0.059)	55.5 vs. 16.1 (*p* < 0.001)	3.7 vs. 1.8 (*p* < 0.001)	7.4 vs. 4.8 (*p* < 0.001)

**Table 2 ijms-24-05840-t002:** Summary of the main ongoing trials of fruquintinib in colorectal cancer.

Trial	Phase	Therapy-Line	Treatment	Primary End-Point
NCT01975077	II	First-line	FOLFOX/FOLFIRI, fruquintinib	ORR
NCT05634590	II	Second-line	FOLFOX/FOLFIRI, fruquintinib	PFS
NCT05555901	II	Second-line	FOLFIRI plus fruquintinib vs. FOLFIRI plus bevacizumab	PFS
NCT05522738	Ib/II	Second-line	FOLFIRI, fruquintinib	ORR
NCT05004831	II	Third-line	Fruquintinib, trifluridine/tipiracil	PFS
NCT05447715	II	Second-/Third-line	Fruquintinib sequential bevacizumab plus FOLFIRI vs. bevacizumab plus FOLFIRI sequential fruquintinib	PFS
NCT04695470	II	Chemo-refractory	Fruquintinib, sintilimab	PFS
NCT04296019, NCT05016869, NCT05451719, NCT04733963, NCT05659290	II or I/II	Mantainance	Fruquintinib or fruquintinib plus capecitabine	PFS
NCT04582981	II	Chemo-refractory	Fruquintinib plus raltitrexed vs. fruquintinib	PFS
NCT04866862	II	Chemo-refractory	Fruquintinib, camrelizumab	ORR

## Data Availability

All data generated or analyzed during this study are included in this published article.

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
