# Peer review of "Evaluation of Fruquintinib in the Continuum of Care of Patients with Colorectal Cancer"

_ijms, 2023, doi:10.3390/ijms24065840_

Round 1
Reviewer 1 Report
I would like to thank authors for the excellent review. Their manuscript summarises nicely the current role of fruquintinib in the management of advanced colon cancer
Minor Reviews:
PDX abbreviation (line 117) will need to be explained
In Table 1 statistical significant difference should be indicated when appropriate. Please also check the commas in the Fresco trial (should be points)
I would welcome the authors to comment on the role of clinical biomarkers (e.g. development of hypertension or HFS) on predicting fruquintinib response.
A second table to summarise the ongoing trials would be really helpful.
Is there any information on resistance mechanisms to fruquintinib?
Where exactly do the authors feel fruquintinib would fit in the continuum of care for the management of CRC patients?
Author Response
I would like to thank authors for the excellent review. Their manuscript summarises nicely the current role of fruquintinib in the management of advanced colon cancer
Minor Reviews:
PDX abbreviation (line 117) will need to be explained
We added “patient-derived xenograft (PDX)” in line 117 and in abbreviation section.
In Table 1 statistical significant difference should be indicated when appropriate. Please also check the commas in the Fresco trial (should be points)
As suggested, we added p values In Table 1 and we pointed the medians and rates of FRESCO trial.
I would welcome the authors to comment on the role of clinical biomarkers (e.g. development of hypertension or HFS) on predicting fruquintinib response. A second table to summarise the ongoing trials would be really helpful. Is there any information on resistance mechanisms to fruquintinib?
We added these sentences focusing on biomarker of response to antiagiogenics
“Every effort should be made to identify clinical and molecular biomarkers predictive of response. Potential factors associated with treatment response to antiangiogenics and multitarget agents included LDH, hERG1/aHIF-2α expression, circulating angiopoietin-2 level, but results should be confirmed in prospective studies. [48 - 52] An exploratory analysis from the CORRECT trial showed an association between HFS reaction and survival benefit from regorafenib and similar differential benefit was observed for fruquintinib in a post-hoc analysis of the FRESCO trial. [53, 54] “
Unfortunately, no resistance mechanism has been successfully demonstrated for fruquintinib.
- Inghilesi, A.L., Gallori, D., Antonuzzo, L., et al. 2014. Predictors of survival in patients with established cirrhosis and hepa-tocellular carcinoma treated with sorafenib. World J. Gastroenterol. WJG 20, 786–794. https://doi.org/10.3748/wjg.v20.i3.786.
- Marmorino, F., Salvatore, L., Barbara, C., et al. 2017. Serum LDH predicts benefit from bevacizumab beyond progression in metastatic colorectal cancer. Br. J. Cancer 116, 318–323. https://doi.org/10.1038/bjc.2016.413.
- Iorio, J., Lastraioli, E., Tofani, L., et al. 2020. hERG1 and HIF-2α Behave as Biomarkers of Positive Response to Bevacizumab in Metastatic Colorectal Cancer Patients. Transl. Oncol. 13, 100740. https://doi.org/10.1016/j.tranon.2020.01.001.
- Antoniotti, C., Marmorino, F., Boccaccino, A., et al. 2022. Early modulation of Angiopoietin-2 plasma levels predicts benefit from regorafenib in patients with metastatic colorectal cancer. Eur. J. Cancer 165, 116–124. https://doi.org/10.1016/j.ejca.2022.01.025.
- Cosso F., Lavacchi D., Fancelli S., et al. Long-term response of more than 9 years to regorafenib in a heavily pretreated patient with metastatic colorectal cancer. Anticancer Drugs. 2023 Mar 1;34(3):451-454. doi: 10.1097/CAD.0000000000001410. Epub 2022 Nov 15. PMID: 36730636.
- Grothey A., Huang L., Wagner A., et al. Hand-foot skin reaction (HFSR) and outcomes in the phase 3 CORRECT trial of regorafenib for metastatic colorectal cancer (mCRC). 10.1200/JCO.2017.35.15_suppl.3551 J Clin Oncol 35, no. 15_suppl (May 20, 2017) 3551-3551.
- Li J., Qin S., Bai Y., et al. Association between hand-foot skin reaction (HFSR) and survival benefit of fruquintinib in FRESCO trial. 10.1200/JCO.2019.37.15_suppl.e15012 J Clin Oncol 37, no. 15_suppl.
We also added Table 2: Summary of the main ongoing trials of fruquintinib in colorectal cancer.
Where exactly do the authors feel fruquintinib would fit in the continuum of care for the management of CRC patients?
Fruquintinib is a suitable option for chemo-refractory disease and in specific for patients who have been previously treated with fluoropyrimidine-, oxaliplatin-, and irinotecan-based chemotherapy, an anti-vascular endothelial growth factor biological therapy, and, if RAS wild-type, an anti-epidermal growth factor receptor EGFR therapy.
Reviewer 2 Report
An interesting and significant article summarizing several studies. The topic is very relevant. Information is presented clearly. The obtained results are important and valid. The discussion is appropriate and provides an appropriate context for understanding and evaluating the results, excluding the paragraph on radiological disease control, as this topic is not supported by any published data from this study.
Author Response
An interesting and significant article summarizing several studies. The topic is very relevant. Information is presented clearly. The obtained results are important and valid. The discussion is appropriate and provides an appropriate context for understanding and evaluating the results, excluding the paragraph on radiological disease control, as this topic is not supported by any published data from this study.
Thanks.
Of course, the radiological considerations are referred to regorafenib. We then specified “To date, little is known about the specific pattern of response of fruquintinib, but less explored radiological parameters, such as cavitation in lung metastases, should be extensively studied as early biomarker of clinical benefit.”
Reviewer 3 Report
The very first statement needs a citation.
The Introduction can benefit from some editing as there are a number small paragraphs that can either be elaborated or merged with other paragraphs. Similarly, see the very next section that starts with 3 small paragraphs. This is also true for subsequent sections – please refrain from writing small paragraphs. Edit them so that the write-up has well-structured paragraphs.
Authors need to provide a cartoon diagram to showcase the biological effects of fruquintinib, as also verified in multiple clinical trials.
Section ‘Brief Discussion and Perspectives’ is definitely not brief. Please come up with another section heading to filter out much of the discussion and add ‘perspectives’ to stand-alone section at the end of manuscript.
Author Response
The very first statement needs a citation.
REF: Siegel R.L., Miller K.D., Fuchs H.E., et al. Cancer statistics, 2022. CA Cancer J Clin. 2022 Jan;72(1):7-33. doi: 10.3322/caac.21708. Epub 2022 Jan 12. PMID: 35020204.
The Introduction can benefit from some editing as there are a number small paragraphs that can either be elaborated or merged with other paragraphs. Similarly, see the very next section that starts with 3 small paragraphs. This is also true for subsequent sections – please refrain from writing small paragraphs. Edit them so that the write-up has well-structured paragraphs.
We restructured the paragraphs based on the comments of the reviewers. We integrated small paragraphs within larger paragraphs.
Authors need to provide a cartoon diagram to showcase the biological effects of fruquintinib, as also verified in multiple clinical trials.
We added Fig 1.
Section ‘Brief Discussion and Perspectives’ is definitely not brief. Please come up with another section heading to filter out much of the discussion and add ‘perspectives’ to stand-alone section at the end of manuscript.
According to reviewer we separated in two sections “brief discussion” and “perspectives”
Round 2
Reviewer 3 Report
Thanks for addressing all of my concerns !!